# BRG1 and NPM-ALK Are Co-Regulated in Anaplastic Large-Cell Lymphoma; BRG1 Is a Potential Therapeutic Target in ALCL

**DOI:** 10.3390/cancers14010151

**Published:** 2021-12-29

**Authors:** Gavin D. Garland, Stephen P. Ducray, Leila Jahangiri, Perla Pucci, G. A. Amos Burke, Jack Monahan, Raymond Lai, Olaf Merkel, Ana-Iris Schiefer, Lukas Kenner, Andrew J. Bannister, Suzanne D. Turner

**Affiliations:** 1Division of Cellular and Molecular Pathology, Department of Pathology, University of Cambridge, Cambridge CB2 0QQ, UK; gdg26@mrc-tox.cam.ac.uk (G.D.G.); ducrays@tcd.ie (S.P.D.); leila.jahangiri@bcu.ac.uk (L.J.); PP504@cam.ac.uk (P.P.); 2Department of Life Sciences, Birmingham City University, Birmingham B15 3TN, UK; 3Department of Paediatric Oncology, Cambridge University Hospital NHS Trust, Cambridge CB5 8PD, UK; Amos.Burke@Addenbrookes.nhs.uk; 4The European Bioinformatics Institute (EMBL EBI), Wellcome Genome Campus, Cambridge CB10 1SA, UK; monahanj@ebi.ac.uk; 5Department of Laboratory Medicine and Pathology, University of Alberta, Edmonton, AB T6G 2R3, Canada; rlai@ualberta.ca; 6Department of Pathology, Medical University Vienna, 1090 Vienna, Austria; Olaf.merkel@meduniwien.ac.at (O.M.); ana-iris.schiefer@meduniwien.ac.at (A.-I.S.); lukas.kenner@meduniwien.ac.at (L.K.); 7Unit of Pathology of Laboratory Animals, University of Veterinary Medicine Vienna, 1210 Vienna, Austria; 8CBMed, 8010 Graz, Austria; 9Christian Doppler Laboratory of Applied Metabolomics (CDL-AM), Medical University Vienna, 1090 Vienna, Austria; 10The Gurdon Institute, Cambridge CB2 1QN, UK; ab149@cam.ac.uk; 11Central European Institute of Technology (CEITEC), Masaryk University, 601 77 Brno, Czech Republic

**Keywords:** NPM-ALK, ALCL, Brg1

## Abstract

**Simple Summary:**

T-cell lymphoma is a cancer of the immune system. One specific sub-type of T-cell lymphoma is a malignancy called anaplastic large cell lymphoma (ALCL), which is distinct from the other forms, as in general, it has a better prognosis. Research conducted to understand why ALCL develops has shown that a specific genetic event occurs, whereby a new protein is created that drives tumour growth. This protein is called nucleophosmin–anaplastic lymphoma kinase (NPM-ALK). Our research, described here, shows that NPM-ALK regulates another protein, called BRG1, to drive proliferation of tumour cells. In turn, when the gene that leads to expression of BRG1 is inactivated, the tumour cells die. These data suggest that therapeutic targeting of BRG1 might be a novel therapy for this form of cancer.

**Abstract:**

Anaplastic large-cell lymphoma (ALCL) is a T-cell malignancy driven in many cases by the product of a chromosomal translocation, nucleophosmin–anaplastic lymphoma kinase (NPM-ALK). NPM-ALK activates a plethora of pathways that drive the hallmarks of cancer, largely signalling pathways normally associated with cytokine and/or T-cell receptor-induced signalling. However, NPM-ALK is also located in the nucleus and its functions in this cellular compartment for the most part remain to be determined. We show that ALCL cell lines and primary patient tumours express the transcriptional activator BRG1 in a NPM-ALK-dependent manner. NPM-ALK regulates expression of BRG1 by post-translational mechanisms dependent on its kinase activity, protecting it from proteasomal degradation. Furthermore, we show that BRG1 drives a transcriptional programme associated with cell cycle progression. In turn, inhibition of BRG1 expression with specific shRNA decreases cell viability, suggesting that it may represent a key therapeutic target for the treatment of ALCL.

## 1. Introduction

Anaplastic large-cell lymphoma (ALCL) is a T cell Non-Hodgkin Lymphoma (NHL) which accounts for 10–15% of paediatric/adolescent cases [1], and which can be separated into two distinct subclasses based on differential expression of anaplastic lymphoma kinase (ALK). Both subclasses share similar histological features, although ALK^+^ ALCL patients are typically children or young adults and have a relatively good prognosis, with 88–95% of patients achieving complete remission [2,3], whereas ALK^-^ ALCL patients are usually 40–65 years of age at diagnosis and have a less favourable prognosis, with a 5 year overall survival of 49% [4].

For the majority of ALK^+^ ALCL cases, expression of ALK is the consequence of the t(2;5)(p23;q35) chromosomal translocation that results in the fusion of ALK with nucleophosmin 1 (NPM), thereby generating the NPM-ALK protein chimaera with strong expression driven by the NPM promoter, and oligomerisation potential that permits constitutive activation of NPM-ALK tyrosine kinase activity [5,6]. The oncogenic potential of NPM-ALK has been well established and is dependent on its tyrosine kinase activity, activating a variety of intracellular signalling pathways involved in cell survival, proliferation and cell cycle progression including the RAS/ERK, PI-3K/AKT, and JAK/STAT pathways [5,7,8,9,10,11,12]. Furthermore, selective inhibition of ALK with small-molecule inhibitors has shown encouraging responses in patients with relapsed, advanced ALK^+^ ALCL [13]. It has been reported that the oncogenic effects of NPM-ALK are mediated in part via epigenetic mechanisms, whereby silencing of CpG islands within promoter regions of proximal T-cell receptor (TCR) signalling proteins and STAT5B have been reported, putatively as a consequence of NPM-ALK-induced expression of DNMT1 [13,14]. However, the molecular mechanisms accounting for transcriptional regulation in ALCL, and the role played by NPM-ALK in mediating these, largely remain to be determined, although STAT3 has been shown to play a key role in ALCL [15,16].

BRM-Related Gene 1 (BRG1, also known as SMARCA4), is a central ATPase subunit of the human SWI/SNF chromatin remodelling complex [17,18]. BRG1 can relieve DNA CpG hypermethylation-induced epigenetic silencing of genes by directly interacting with their promoter in a manner that is dependent on the enzyme’s ATPase activity, thereby inducing transcriptional activation [19]. Previous studies have shown that BRG1 immunoprecipitates with an amino-terminal NPM1 antibody in the NPM-ALK^+^ ALCL cell line KARPAS-299, suggesting that it is part of a complex with NPM-ALK (NPM1 also binds to NPM-ALK via the oligomerisation domain of NPM retained in the fusion protein which allows its nuclear translocation) [14]. Whilst BRG1 has classically been seen as a tumour suppressor gene, with its loss promoting tumourigenesis in vivo, it has been shown that BRG1 can act in oncogenic roles in some cancers [20,21,22,23,24]. Indeed, we find that expression of BRG1 and NPM-ALK proteins is co-regulated. BRG1 expression is mediated by NPM-ALK, at least in part through protection from proteasomal degradation. Furthermore, BRG1 promotes the transcription of genes involved in cell cycle regulation in ALCL cell lines and its genetic inhibition decreases cell viability. At present, specific clinically actionable inhibitors of BRG1 are not available; further, should they become available in the future, BRG1 activity might be inhibited for therapeutic purposes.

## 2. Materials and Methods

### 2.1. Cell Lines

Cell lines Jurkat, SUDHL-1, Karpas-299, DEL and SUP-M2 were purchased from DSMZ (Braunschwieg, Germany). The ALCL, ALK- cell line, FEPD was a kind gift provided by Prof Annarosa Del Mistro, University of Padua, Italy. All of the cell lines were cultured in RPMI-1640/10% FBS (Gibco, Loughborough, UK/Biosera, Heathfield, UK) supplemented with 1% Penicillin-Streptomycin (Gibco, Loughborough, UK), in an incubator at 37 °C/5% CO_2_. The DHL-T5 A5 (a derivative SUDHL1 cell line expressing an shRNA targeting NPM-ALK in a doxycycline-inducible manner) and MEFNA (murine embryonic fibroblast cell line expressing NPM-ALK in a doxycycline-inducible manner) cell lines were provided by Roberto Chiarle, University of Torino, Torino, Italy. For induction of shRNA expression, the DHL TA A5 cell line was grown in the presence of 1 µg/mL doxycycline (Sigma-Aldrich, Poole, UK). To inhibit expression of NPM-ALK, the MEFNA cell line was grown in the presence of 1 µg/mL doxycycline (Sigma-Aldrich, Poole, UK). A tumour cell line derived from the CD4/NPM-ALK transgenic mouse model [11] (CD4-NA cell line) was kindly provided by Roberto Chiarle, University of Torino, Torino, Italy.

### 2.2. Inhibitors and Antibodies

The ALK/c-MET inhibitor Crizotinib was purchased from LC Laboratories (Woburn, MA, USA) and was resuspended in DMSO (Sigma-Aldrich, Poole, UK) to a stock concentration of 1 mM, before dilution in DMSO (Sigma-Aldrich) or sterile Dulbecco’s PBS (PAA, Pasching, Austria) to the required concentration for each assay as indicated. Brigatinib and lorlatinib were purchased from MedChem Express (Monmouth Junction, NJ, USA). The BRG1 antibody for IHC was purchased from Abcam (Cambridge, UK). Western blot antibodies were purchased from Cell Signaling Technologies (Danvers, MA, USA) (pALK Y1278, pALK Y1604, STAT3 cleaved caspase 3, p53, p21), Life Technologies (Waltham, MA, USA) (ALK), Abcam (Cambridge, UK) (Brg1) and Sigma (Poole, UK) (Tubulin).

### 2.3. Primary Patient Samples and Immunohistochemistry

Immunohistochemistry was conducted as described previously [25]. In brief, a tissue microarray containing cores of 12 ALK+ ALCL and 9 ALK- ALCL FFPE patient samples as well as 8 Peripheral T-cell Lymphoma—Not Otherwise Specified (PTCL-NOS) and 6 Angioimmunoblastic T-cell Lymphoma (AITL) patient samples was subjected to antigen retrieval following incubation in a citrate buffer before exposure to BRG1 antibody (Abcam, Cambridge, UK) diluted 1:100 in PBS/BSA overnight at 4 °C. Tumours were defined as being highly positive for BRG1 (++) if >30% cells stained highly positive and BRG1 + if >30% tumour cells stained positive for BRG1.

### 2.4. Mice

Mice expressing NPM-ALK from the CD4 promoter [11] were kindly provided by Roberto Chiarle (University of Torino, Torino, Italy) and were housed as previously described [15].

### 2.5. SDS-PAGE and Western Blot

SDS-PAGE and Western blot were conducted as described previously [16]. Briefly, cells (0.5–5.0 × 10^6^) were lysed in RIPA buffer, mixed with 2 × Laemmli’s loading buffer before boiling and loading into the wells of an 8% SDS-PAGE. Proteins were transferred to Immobilon PVDF membranes (Bio-Rad, Watford, UK) by wet transfer before incubation in 5% (*w*/*v*) BSA (PAA, Pasching, Austria) for 1 h at RT followed by primary antibody for 1 h at RT or overnight at 4 °C (phospho-specific antibodies), washed twice in TBS/T and then incubated with horse radish-peroxidase (HRP)-conjugated secondary antibody at 1:10,000 for 1 h at RT. The membrane was subsequently washed twice in TBS/T, and then exposed to Immobilon Western chemiluminescent HRP substrate (Millipore, Watford, UK) and imaged using a Fujifilm LAS-4000 Biomolecular Imager (Raytek, Sheffield, UK). Analysis and densitometry were performed using Aida software (Raytek, UK).

### 2.6. RNAseq

RNAseq was conducted by Novogene, Cambridge, UK. Data were mapped with HISAT2 [17], quantified (produced count tables) with htseq-count [18] and analysed in R using various Bioconductor packages (e.g., DESeq2, gplots, and RColorBrewer; http://colorbrewer2.org, accessed on 31 July 2021) [19]. This pipeline was broken down into: initial set-up (building an index from the GRCh38 primary assembly with the corresponding transcriptome annotation from Ensembl version 99) [20], reading mapping with HISAT2, gene expression quantification with HTSeq, data normalisation, post-normalisation QC, normalisation of counts with DESeq2’s variance stabilising transformation (VST) for visualisation purposes, principal component analysis (PCA) of sample to sample variation, detection of statistically significant (*p*-value ≤ 0.05, *p*-value ≤ 0.005, or adj. *p*-value ≤ 0.05 are outlined and specified where appropriate) and differentially expressed genes using DESeq2’s negative binomial Wald test, and post-statistical QC steps in R.

### 2.7. Isolation of RNA

RNA was isolated using the RNeasy Plus mini kit (Qiagen, Manchester, UK) according to the manufacturer’s instructions. Briefly, 0.5–4.0 × 10^6^ cells were lysed in buffer RLT plus and passed through an RNeasy spin column. Subsequently, RNA was eluted in DEPC-treated water and the concentration was measured by NanoDrop (Thermo Fisher Scientific, Waltham, MA, USA). Sufficient RNA purity was indicated by A260/280 ratios of ~2.0 and A260/230 ratios of ~2.0–2.2. To completely remove contaminating DNA, 10 µg purified RNA was diluted in 50 µL 1× Turbo DNase Buffer (Life Technologies, Waltham, MA, USA) supplemented with 2U Turbo DNase (Life Technologies, Waltham, MA, USA) and incubated at 37 °C for 30 min. The DNase was inactivated by the addition of 7.5 µL 100 mM EDTA, pH 8.0.

### 2.8. Synthesis of cDNA

Synthesis of cDNA was performed using the SuperScript II Reverse Transcriptase kit (Life Technologies, Waltham, MA, USA) according to the manufacturer’s protocol. Briefly, 500 ng DNase-treated RNA was diluted in DEPC-treated water supplemented with 500 ng oligo(dT) primers (Stratagene, San Diego, CA, USA) and 10 nmol dNTP mix (Fermentas, Waltham, MA, USA) up to 12 µL total volume and incubated at 65 °C for 5 min. To this, 4 µL 5× First Strand Buffer and 2 µL 0.1 M DTT were added and then incubated at 42 °C for 2 min. Subsequently, 1 µL SuperScript II Reverse Transcriptase was added and incubated at 42 °C for 50 min to synthesise the cDNA. The reaction was inactivated by incubation at 70 °C for 15 min. The cDNA concentration was measured by NanoDrop analysis (Thermo Fisher Scientific, Waltham, MA, USA). Sufficient cDNA purity was indicated by A260/280 ratios of ~1.8 and A260/ratios of ~2.0–2.2.

### 2.9. Quantitative Real-Time PCR Analysis

Samples of cDNA were analysed by quantitative, real-time (qRT)-PCR using the QuantStudio 6 Flex Real-Time PCR system (Thermofisher, Waltham, MA, UK). Reactions were performed in 10 µL volumes with 40 ng cDNA, 50% (*v*/*v*) SsoFast EvaGreen Supermix (Bio-Rad, Watford, UK) and 25 ng of each primer. The primers and conditions used are detailed in Table 1. Each reaction was performed in triplicate and data points with a standard deviation >1.0 were excluded. These data were analysed using the comparative Ct method. Alternatively, the PCR efficiency was calculated from a 5-point, 4-fold dilution series from 160 ng to 0.625 ng cDNA prepared from the SU-DHL-1 or JURKAT cell line using a QIAquick PCR purification kit (Qiagen, Manchester, UK) and eluted in DEPC-treated water, and the data were then analysed using the absolute quantification method. The correlation co-efficient of the calibration curves was >0.95. These data were analysed using Quantstudio software (Illumina, Cambridge, UK) and were transferred to Microsoft Excel 365 (Microsoft, Redmond, WA, USA) and GraphPad Prism 5 (GraphPad Software, San Diego, CA, USA) for further analysis. Data were analysed using two-tailed, unpaired *t*-tests (assuming equal variance) on GraphPad Prism 5 with a significance level set at *p* < 0.05.

### 2.10. Lentiviral Transduction of shRNA

HEK293FT cells were seeded the day before transfection with the intent to be 80–90% confluent on the day of transfection. The lentiviral construct was transfected at a 1:1:1 ratio with the packaging plasmids psPax2 and pMD2.G with TransIT-292 (MirusBio, Madison, WI, USA) in OptiMEM (ThermoFisher, Waltham, MA, USA) into the HEK293FT cells. Viral particles were harvested 60 h following transfection and filtered through a 0.45 μm PVDF membrane and either used immediately or stored at −80 °C. Cells were seeded at a concentration of 1 million cells/mL in 1 mL of medium in the wells of a 6-well plate. Dropwise, 500 µL of room temperature-thawed viral particle-containing medium was added to the cells and, 24 h later, transduced cells were selected in puromycin at 1 µg/mL. Lysates were prepared from SUDH1 and FEPD cells 48 and 72 h post-puromycin selection following scrambled and BRG1 shRNA transfection (Table 2) and analysed for expression of the key cell cycle regulators p53, p21, and cleaved caspase 3.

### 2.11. Viability and Caspase Assays

CD4-NA cells (10^4^) were seeded in 96-well white polystyrene microplates (Corning, Waltham, MA, USA) and transfected with BRG1-targeting siRNAs (10 nM). After 6 days, 20 µL CellTiter-Blue cell viability reagent (Promega, Madison, WI, USA) was added to each well and cells were incubated at 37 °C for 4 h, according to the manufacturer’s protocol. Fluorescence was read on a SpectraMax i3 microplate reader (Molecular Devices, San Jose, CA, USA). For assessment of caspase 3/7 activity, the cells were instead incubated with 100 µL of Caspase 3/7-Glo apoptosis reagent (Promega, Madison, WI, USA) for 1.5 h. Luminescence was read on a SpectraMax i3 microplate reader (Molecular Devices, San Jose, CA, USA). Data were normalised to the scrambled non-coding siRNA.

## 3. Results

### 3.1. BRG1 Is Expressed in ALK^+^ ALCL and Other Peripheral T-Cell Lymphomas

Western blots performed on protein lysates show that all T-cell lymphoma cell lines examined (including ALK^+^ ALCL cell lines) express BRG1 at varying levels, whereas the acute T-cell leukaemia cell line JURKAT expresses BRG1 at higher levels (Figure 1a).

Furthermore, Western blot analysis of protein lysates prepared from thymic tumours obtained from the CD4/NPM-ALK transgenic mouse line [11,15] also express BRG1 albeit at a lower level than that observed in ALK^+^ ALCL cell lines (Figure 1a). Immunohistochemical (IHC) analysis of a tissue microarray panel of T-cell lymphomas composed of primary tissue biopsies were also positive for BRG1 expression in all ALK^+^ ALCL patients analysed, with the majority showing strong expression (present in >30% of tumour cells; 12/12 cases positive: 2 cases +, 10 cases ++ staining) (Figure 1b–d). Indeed, this was also the case for ALK^-^ ALCL (8/9 cases positive: 3 cases +, 5 cases ++), PTCL-NOS (8/8 cases positive, all ++) and AITL (6/6 cases positive: 2 cases +, 4 cases ++), with only one ALK^-^ ALCL patient showing negative staining for BRG1.

### 3.2. NPM-ALK Regulates Expression of BRG1 by Protecting It from Proteasomal Degradation

To examine whether BRG1 might be regulated by NPM-ALK tyrosine kinase activity in ALK^+^ ALCL cell lines, SU-DHL-1 cells were treated with the ALK inhibitor Crizotinib for 6 h (Figure 2a).

Western blots performed on protein cell lysates show that Crizotinib treatment depletes the levels of NPM-ALK pY338 in these cells and also causes a concomitant reduction in BRG1 (Figure 2a). To confirm this result, the modified SU-DHL-1 cell line DHL TA A5, which expresses a doxycycline-inducible shRNA against NPM-ALK, was used [21]. Treatment of these cells for 72 or 96 h in the presence of doxycycline resulted in depletion of total NPM-ALK and NPM-ALK (pY664) protein levels and a concomitant downregulation of BRG1 (Figure 2b). Finally, the modified mouse embryonic fibroblast cell line MEFNA, in which human NPM-ALK is expressed under the transcriptional control of the Tet-On gene expression system [22] was grown in the presence of doxycycline resulting in the induction of NPM-ALK and NPM-ALK pY664 and consequently BRG1 up-regulation (Figure 2c). In all, these data suggest that NPM-ALK controls expression of BRG1 most likely through an intermediary protein(s).

Having established the potential (in)direct regulation of BRG1 expression by NPM-ALK, we sought to uncover the mechanism of this activity. Transcript levels of BRG1 were assessed following inhibition of NPM-ALK activity with the ALK tyrosine kinase inhibitors lorlatinib, brigatinib and crizotinib. Following 6 h of incubation, no significant change in transcript levels were observed, suggesting a post-translational mechanism of regulation (Figure 3a). In support of this, inhibition of NPM-ALK expression in 2 independent cell lines via doxycycline-inducible shRNA expression also did not lead to a significant decrease in BRG1 transcript levels (Figure 3b). However, co-treatment of cells with an ALK inhibitor and the proteasomal inhibitor bortezomib maintained BRG1 expression levels confirming that NPM-ALK regulates expression of BRG1 through the proteasome rather than at the transcriptional level (Figure 3c).

### 3.3. BRG1 Induces Transcription of Genes Involved in Cell Cycle Progression and Its Inhibition Results in a Loss of Cell Viability

To determine the consequences of Brg1 expression in ALCL, cells were transfected with shRNA to Brg1 leading to a decrease in Brg1 transcript and protein levels for three of the four shRNA tested (Figure 4a,b). RNA was isolated 72 h later from the shRNA5-expressing cells and sequencing conducted to examine changes to the transcriptome mediated by Brg1 (Figure 4c). Following differential gene expression analysis of the scrambled shRNA versus shRNA5, the transcriptome was identified as being dramatically altered showing both downregulation (2527 mRNA transcripts: of a cumulative 26,833 genes, adj. *p* ≤ 0.05) and upregulation (2900 mRNA transcripts, of a cumulative 26,833 genes, adj. *p* ≤ 0.05) of genes (Table 1, Figure 4c). When analysing the 15,285 protein-coding genes alone, 1934 (13%) were downregulated while 2797 (18%) were upregulated (adj. *p* ≤ 0.05). To establish which physiological processes are most affected by the activity of BRG1, pathway enrichment analyses were undertaken. The input comprised of the significant (adj. *p* ≤ 0.05) protein-coding targets, where the absolute log2-fold change in expression was ±1.5. Transcripts whose expression was either induced (*n* = 33) or repressed (*n* = 243) by BRG1 were investigated. Reactome analysis suggests that BRG1 plays roles in upregulating genes associated with the cell cycle (Table 3), while suppressing genes associated with epigenetic regulation (Table 4). Genes associated with cell cycle regulation were validated by RT-PCR including *CDC6*, *E2F2* and *RPM2* (Figure 4d), suggesting that Brg1 activity in ALCL plays a role in regulating cell cycle progression. Given this finding, we assessed the role of BRG1 in maintaining cell viability.

Brg1 expression was inhibited following transduction of cells with 3 independent shRNA targeting Brg1 for 14 days. In all three cases, cell viability was significantly reduced compared to cells transduced with a non-targeting shRNA control vector. Comparing the relative viable cell number between scrambled shRNA and BRG1-targeted shRNA cultures after 14 days showed shRNA2 transduction resulted in a 78% decrease in relative viable cell number while shRNA3 and shRNA5 transduction resulted in a 40% and 79% decrease, respectively (Figure 4e).

To determine the mechanism of reduced cell viability, expression of p53, p21 and cleaved caspase 3 was assessed in SUDHL1 and FEPD cells at 48 and 72 h post-puromycin selection following shRNA-mediated knockdown of BRG1 (Appendix A). However, this revealed no significant changes in expression levels, suggesting that, at these time points, neither cell cycle arrest induced by p21 nor apoptosis as determined by cleaved caspase 3 are responsible for reduced cell viability. Similarly, cleaved caspase 3 levels were unchanged in CD4-NA mouse tumour cell lines treated with siRNA-targeting BRG1 (Appendix A). Collectively, these observations suggest that the reduction in viability of BRG1 knock-down cells may not be linked to cell cycle arrest mediated by p21 nor apoptosis at the time points tested.

## 4. Discussion

Loss of BRG1 has been associated with various cancers [23,24,26,27], and has been shown to promote tumourigenesis in vivo [28,29,30]. Hence, Brg1 has been classically viewed as a tumour suppressor gene. However, Brg1 is typically expressed in all cell types of adult tissue that undergo proliferation or self-renewal [31], so it is not necessarily surprising that human T-cell lymphomas (including ALK^+^ ALCL) express Brg1. Indeed, recent evidence indicates that BRG1 can fulfil pro-oncogenic roles in some cancers [32]. It is therefore possible that BRG1 interacts (in)directly with NPM-ALK via its hetero-oligomerisation partner NPM1, which warrants further investigation. In support of this, it has already been reported that Brg1 forms a complex with NPM-ALK as it has been detected by mass spectrometry of NPM1 immunoprecipitates in an ALCL cell line [14] and NPM1 has been identified by mass spectrometry in anti-FLAG immunoprecipitates prepared from nuclear extracts of 293T cells that ectopically express FLAG-Brg1 [33].

We also show that Brg1 expression is positively regulated by NPM-ALK tyrosine kinase activity in ALK^+^ ALCL cell lines at a post-translational level. Specifically, NPM-ALK, in a tyrosine kinase-dependent manner protects Brg1 from proteasomal degradation although the exact mechanism remains to be determined. Previously, others have shown that the ALK receptor, post-activation, recruits the Cbl ubiquitin ligase [34]. ALK activation by agonist monoclonal antibody (mAb) stimulation led to a significant increase in ALK ubiquitinoylation by receptor phosphorylation and the subsequent recruitment of Cbl [35]. Furthermore, NPM-ALK has been shown to bind to and phosphorylate STAT1, thereby promoting its proteasomal degradation [36], and carboxyl hsp70-interacting ubiquitin ligase increases the degradation of NPM-ALK [37]. Collectively, this evidence highlights the interaction of NPM-ALK with the proteasomal degradation machinery which may likewise be activated in the regulation of Brg1.

Finally, we demonstrate that BRG1 regulates the transcriptome of ALCL to promote the expression of genes involved in the cell cycle and to maintain cell viability. BRG1 knockdown led to reduced viability; however, the mechanism involved remains to be further substantiated. Notably, a reduction in cell viability was not significant until 14 days following knockdown of BRG1, suggesting that this is a mild effect that might not be therapeutically applicable unless combined with other agents. This clearly requires further investigation should effective and specific pharmacological inhibitors of BRG1 activity be developed. Naturally, the therapeutic efficacy of BRG1 inhibitors would also be dependent on the potential side effects of inhibiting this ubiquitously expressed protein.

## 5. Conclusions

In conclusion, we have demonstrated that BRG1 is expressed in a range of T-cell lymphomas and its protein levels are maintained by NPM-ALK activity in ALK^+^ ALCL cell lines. Furthermore, we show that BRG1 drives transcription of genes associated with cell cycle progression and in evidence of this, inhibition of BRG1 expression results in a decrease in cell viability. Overall, these data suggest that NPM-ALK is a key mediator of BRG1 activity in ALCL.

## Figures and Tables

**Figure 1 cancers-14-00151-f001:**
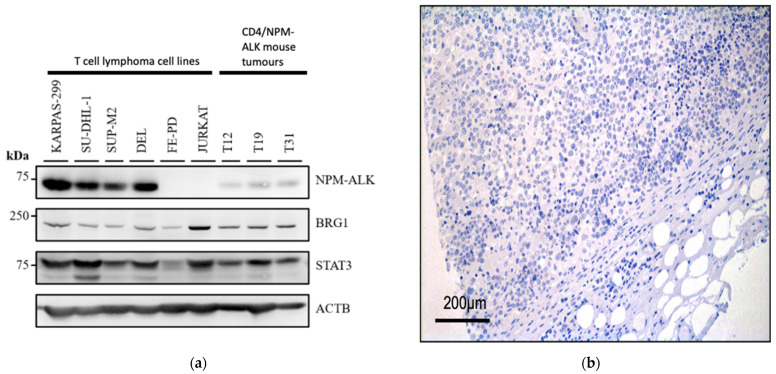
BRG1 is expressed in peripheral T-cell lymphoma. Western blot analysis of the indicated proteins including BRG1 in human ALK^+^ ALCL (KARPAS-299, SU-DHL-1, SUP-M2 and DEL), ALK^−^ ALCL (FE-PD) and acute T-cell leukaemia (JURKAT) cell lines as well as primary tumours from CD4/NPM-ALK transgenic mice (**a**). Representative immunohistochemistry for BRG1 in primary patient tumours of ALCL: (**b**) negative for BRG1 expression, (**c**) positive (+) and (**d**) highly positive staining (++). ACTB = Actin B.

**Figure 2 cancers-14-00151-f002:**
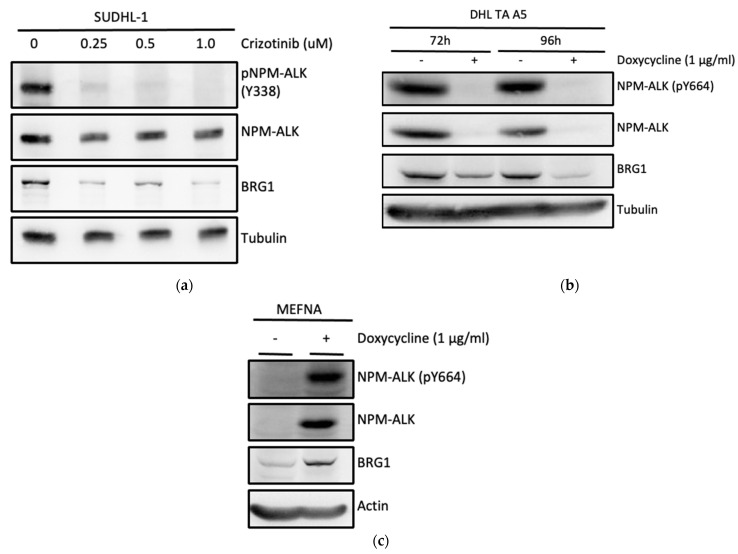
Expression of Brg1 is dependent on NPM-ALK activity in ALCL, ALK+. (**a**) Expression of the indicated proteins analysed by Western blot in SU-DHL-1 cells treated with Crizotinib for 6 h. These data are representative of biological duplicates and data obtained from both the DEL and Karpas-299 cell lines. (**b**) Doxycycline-inducible shRNA-mediated inhibition of NPM-ALK expression in DHL TA A5 cells (derived from the SUDHL-1 cell line) and Western blot analysis of the indicated proteins 72 and 96 h following administration of doxycycline. These data are representative of biological triplicates. (**c**) Induction of Brg1 expression in the MEFNA cell line following doxycycline-induced NPM-ALK expression. These data are representative of biological triplicates.

**Figure 3 cancers-14-00151-f003:**
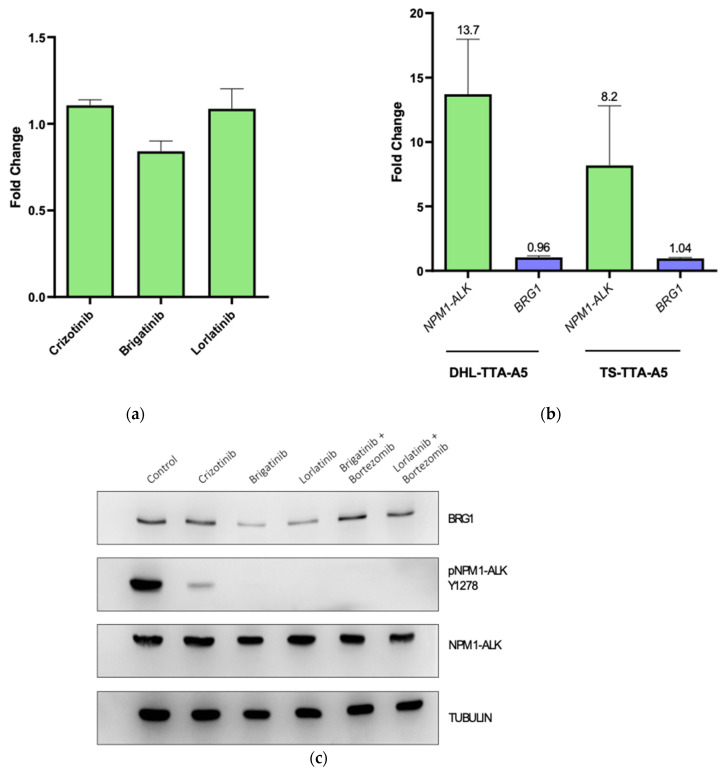
NPM-ALK protects BRG1 from proteasomal degradation. (**a**) BRG1 transcript levels in SU-DHL-1 cells treated for 6 h with the indicated ALK tyrosine kinase inhibitors (each at 200 nM). (**b**) BRG1 mRNA transcript levels in DHL TTA A5 and TS TTA A5 (SUDHL-1 and SUP-M2 derived doxycycline inducible NPM1-ALK shRNA stable cell lines, respectively) following NPM-ALK shRNA induction for 96 h. Data represent the means and standard deviations of biological triplicates. (**c**) Western blot of the indicated proteins following treatment of SU-DHL-1 for 6 h with various ALK inhibitors (crizotinib, brigatinib, or lorlatinib at 200 nM) and brigatinib or lorlatinib with the proteasomal inhibitor bortezomib at 200 nM. Data represent biological triplicates.

**Figure 4 cancers-14-00151-f004:**
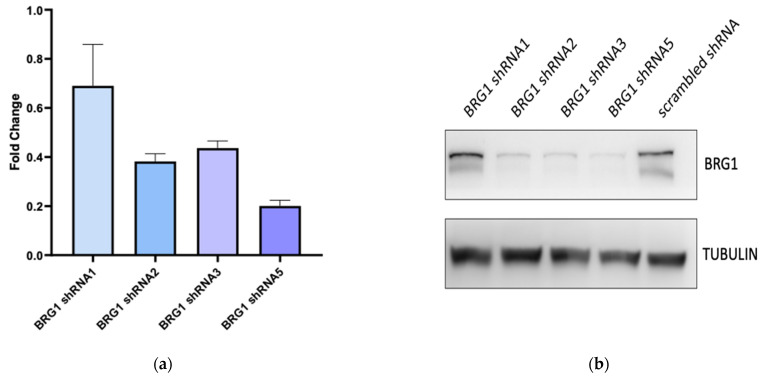
Inhibition of expression of BRG1 in ALCL cell lines alters the transcriptome inducing transcription of genes associated with the cell cycle. (**a**) Transcript levels and (**b**) Western blot for the indicated proteins following knockdown of BRG1 for 72 h by four independent shRNAs (shRNA1; shRNA2; TRCN0000015549, shRNA3; TRCN0000015550 and shRNA5; TRCN0000015551, respectively) versus a scrambled shRNA control (Addgene_1864). Data are representative of biological duplicates. (**c**) Volcano plot of SU-DHL-1 cells following knockdown of BRG1 (SMARCA4) by shRNA5 versus a scrambled shRNA control. The line bisecting the *y*-axis at 1.3010 corresponds to *p* ≤ 0.05 and was utilised as the cut-off for downstream analyses. Differential expression cuts-offs (bisecting *x*-axis at ± 1.5) were set at log2-fold change ± 1.5. Data are interpreted to understand the significance of BRG1; viz. positive expression represents the gene being transcribed in the presence of expressed BRG1 (SMARCA4). Data are representative of biological triplicates (**d**) Transcript levels of the indicated genes in the SU-DHL-1 cell line 72 h following transduction of shRNA5. Data represent the means and standard deviations of biological triplicates. Relative fold changes are illustrated. (**e**) Cell viability as assessed by trypan blue exclusion following BRG1 knockdown by three independent shRNAs v.s. a scrambled shRNA control following 14 days of incubation. Data represent the means and standard deviations of biological triplicates, * *p* < 0.05, ** *p* < 0.005.

**Table 1 cancers-14-00151-t001:** Primers.

Primer	Sequence (5′ to 3′)
BRG1 Forward	TGCTGCGGCCCTTCTTGCTC
BRG1 Reverse	GGTGCCGCCTTTGCCCTTCT
CDC6 Forward	ACCTATGCAACACTCCCCATT
CDC6 Reverse	TGGCTAGTTCTCTTTTGCTAGGA
CLSPN Forward	AAGACAGTGATTCCGAAACAGAG
CLSPN Reverse	TGCGCTTCAAGATTTTCCTGA
E2F2 Forward	CGTCCCTGAGTTCCCAACC
E2F2 Reverse	GCGAAGTGTCATACCGAGTCTT
MYBL2 Forward	CCGGAGCAGAGGGATAGCA
MYBL2 Reverse	CAGTGCGGTTAGGGAAGTGG
PKYMT2 Forward	GCCTGCCAACATCTTCCTG
PKYMT2 Reverse	CCCAGACTGAACACATCCGC
RRM2 Forward	CACGGAGCCGAAAACTAAAGC
RRM2 Reverse	TCTGCCTTCTTATACATCTGCCA
NPM-ALK Forward	CTGTACAGCCAACGGTTTCCC
NPM-ALK Reverse	GGCCCAGACCCGAATGAGG
GAPDH Forward	CCACTCCTCCACCTTTGAC
GAPDH Reverse	ACCCTGTTGCTGTAGCCA
Mouse GAPDH Forward	CATCACTGCCACCCAGAAGACTG
Mouse GAPDH Reverse	ATGCCAGTGAGCTTCCCGTTCAG
Mouse BRG1 Forward	GAAAGTGGCTCTGAAGAGGAGG
Mouse BRG1 Reverse	TCCACCTCAGAGACATCATCGC
Mouse HPRT Forward	CTGGTGAAAAGGACCTCTCGAAG
Mouse HPRT Reverse	CCAGTTTCACTAATGACACAAACG

**Table 2 cancers-14-00151-t002:** shRNA sequences targeting *Brg1*.

Brg1 shRNA	TRC Construct	Sequence
1	TRCN0000015548	CCATATTTATACAGCAGAGAACTCGAGTTCTCTGCTGTATAAATATGG
2	TRCN0000015549	CCGGCCCGTGGACTTCAAGAAGATACTCGAGTATCTTCTTGAAGTCCACGGG
3	TRCN0000015550	CCGGGCCAAGCAAGATGTCGATGATCTCGAGATCATCGACATCTTGCTTGGCTTTTT
5	TRCN0000015552	CCGGCGGCAGACACTGTGATCATTTCTCGAGAAATGATCACAGTGTCTGCCGTTTTT

**Table 3 cancers-14-00151-t003:** Output of Reactome analysis—genes upregulated by BRG1 activity. Genes were input (using Ensembl gene ID) and data were analysed, whereby the most enriched, significant (adj. *p* ≤ 0.05) gene sets (pathways) were identified.

Pathway	Gene Count	Entities Total
Cell cycle	12	734
Cell cycle, mitotic	10	596
Cell cycle checkpoints	6	280
Mitotic G1 phase and G1/S transition	8	173
G2/M checkpoints	5	154
G1/S checkpoints	5	150
DNA replication	3	142
DNA replication pre-initiation	3	88
Apoptotic execution phase	3	54
Transcriptional regulation by E2F6	2	46

**Table 4 cancers-14-00151-t004:** Output of Reactome analysis—genes suppressed by BRG1 activity. Genes were input (using Ensembl gene ID) and data were analysed, whereby the most enriched, significant (adj. *p* ≤ 0.05) gene sets (pathways) were identified.

Pathway	Gene Count	Entities Total
Reproduction	8	123
HDAC deacetylase histones	7	67
Amyloid fibre formation	7	88
Meiosis	7	92
RNA polymerase I promoter opening	6	34
DNA methylation	6	36
PRC2 methylates histones and DNA	6	44
SIRT1 negatively regulated rRNA expression	6	45
ERCC6 (CSB) and EHMT2 (G9a) positively regulate rRNA expression	6	48
Activated PKN1 stimulates transcription of Androgen Receptor (AR) regulated genes KLK2 and KLK3	6	49

## Data Availability

Data available in a publicly accessible repository that does not issue DOIs. This data can be found here: https://zenodo.org/record/5137843#.Ycs3iS-l0dV accessed on 31 July 2021.

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
