# Peer review of "BRG1 and NPM-ALK Are Co-Regulated in Anaplastic Large-Cell Lymphoma; BRG1 Is a Potential Therapeutic Target in ALCL"

_cancers, 2021, doi:10.3390/cancers14010151_

Round 1
Reviewer 1 Report
The manuscript seems to be revised appropriately.
Minor correction:
" Western blots performed on protein cell lysates show that Crizotinib treatment depletes the levels of NPM-ALK pY338 in these cells in a dose-dependent manner and also causes a concomitant reduction in BRG1 (Figure 2a)." -> the deletion is not applied to the main manuscript. Please check this again.
Reviewer 2 Report
The authors addressed all concerns and questions of my previous review to the fullest. Several additional experiments were performed and incorporated to the manuscript, resulting in major improvement of the story, giving the new insights into the BRG1-NPM-ALK interaction.
This manuscript is a resubmission of an earlier submission. The following is a list of the peer review reports and author responses from that submission.
Round 1
Reviewer 1 Report
In the research, the authors try to show that NPM-ALK regulates BRG1 expression in ALCL at post transcriptional levels by protecting proteosomal degradation.
[Major comment]
From the data, authors state that the “data suggest that NPM-ALK controls expression of BRG1” and “ NPM-ALK regulates expression of BRG1 through the proteasome”
However, most of data seem to suggest not directly but indirectly the correlation between NPM-ALK and BRG1. Why would the protecting of proteasome degradation by NPM-ALK specifically induce BRG1 expression, among other various proteins? In the article, it does not seem to be clearly explained about the specific, direct interaction between NPM-ALK and BRG1. Why do the authors focus on BRG1 molecule in association with ALCL of NPM-ALK? If there are any preliminary data of the authors, or previously reported data from others, such background should be more clearly explained and then investigated.
[Minor comment]
Line 199: “ BRG1 at a similar level,~ (Figure 1a)” : I don't know they reveal similar levels. In some cell lines (SU-DHL-1, SUP-M2), protein expression of BRG1 looks weaker than others.
Line 228-229: “ ~ in a dose-dependent manner ~concomitant reduction in BRG1 (Figure 2a) “ : I don't see dose-dependent manner, although reduced expression of pY338 and BRG1 is noted when treated with Crizotinib, regardless of dose. The author should more clearly stated the results and interpretation for Fig 2a.
Line 303: “~72 hours of induction of shRNA5” : Does “Induction” means “transducton”?
Reviewer 2 Report
Garland et al. show here BRG1 being expressed in a range of T-cell lymphomas and that its protein levels are associated with NPM-ALK activity. Furthermore, they state BRG1 as a potential new therapeutic target in ALK+ ALCL, showing reduced cell counts in consequence of BRG1 inhibition by shRNA technology.
The topic is of high interest, as novel therapeutic targets are urgently needed in (ALK+) ALCL. The presentation of the study is clear and well structured. While the data demonstrating an association of NPM-ALK and BRG1 is weak, the high expression of BRG1 in T cell lymphomas and its inhibition demonstrating reduced cell counts, suggesting BRG1 as novel therapeutic target is very exciting. By gene expression analysis, the study provides a potential underlying mechanism for efficiency of BRG1 inhibition in ALK+ ALCL.
There are two major aspects, which need to be improved:
- The results shown here do not fully support the statement that NPM-ALK regulates BRG1 protein levels. The results here show that BRG1 protein levels are associated with NPM-ALK activity. It is known that BRG1 is able to interact with NPM in the nucleus. The authors also mention this interaction in their discussion, based on reports using mass spectrometry. Hetero-oligomerisation of NPM-ALK with wild type NPM – being responsible for ist nuclear localization- might play a role here.
As BRG1 is highly expressed in both ALK+ and ALK- T cell lymphomas, as nicely shown in this study, the BRG1 expression does not seem to be NPM-ALK specific. Therefore, it is rather unlikely that NPM-ALK is a major regulator of BRG1. The posttranslational regulation of BRG1 by NPM-ALK – as the authors state here – might be an indirect effect, potentially via FBW7, which is known to mediate BRG1 proteasomal degradation.
Could the authors discuss these points? Do the authors think, that NPM-ALK regulates BRG1 and this effect is independent of wt NPM?
In Figure 3c, Crizotinib does not seem to reduce BRG1 protein levels although reducing NPM-ALK phosphorylation, as shown in Figure 2A. Could the authors state on that?
The authors should either state down their conclusion, that NPM-ALK regulates BRG1 protein levels or strengthen their statement, for example by Co-IPs demonstrating direct binding – although this could not clarify the exact role of NPM-ALK vs wt NPM in BRG1 regulation.
- The authors nicely show the overexpression of BRG1 in several T cell lymphomas. Furthermore, they investigate the underlying mechanism of reduced cell counts upon BRG1 inhibition by gene expression analysis and detect several associated genes involved in cell cycle regulation.
The authors should strengthen this point by adding functional data such as BRDU/ EdU and Annexin V staining, to further differentiate between cell cycle defects and apoptosis/ viability upon BRG1 inhibition.
Also, it would be interesting to see if BRG1 inhibition has an effect in ALCL-like mouse tumors (such as CD4/ NPM-ALK). This could be investigated either in vivo (transplantation) or ex vivo (such as methylcellulose assay) upon BRG1 inhibition in tumor cells.
As BRG1 is highly expressed not only in ALK+ ALCL, it would be of high interest to investigate, if BRG1 inhibition has an equal effect on ALK- ALCL. The authors could investigate this by using their different ALK+ and ALK- cell lines.